# Effects of Anthocyanin Supplementation on Reduction of Obesity Criteria: A Systematic Review and Meta-Analysis of Randomized Controlled Trials

**DOI:** 10.3390/nu13062121

**Published:** 2021-06-21

**Authors:** Seongmin Park, Munji Choi, Myoungsook Lee

**Affiliations:** 1Department of Food and Nutrition, Sungshin Women’s University, Seoul 01133, Korea; kpsm3003@naver.com (S.P.); answlek@naver.com (M.C.); 2Research Institute of Obesity Sciences, Sungshin Women’s University, Seoul 01133, Korea; 3Hazardous Substances Analysis Division, Gyeongin Regional Office of Food and Drug Safety, Ministry of Food and Drug Safety, Incheon 22133, Korea; 4National Food Safety Information Service, Seoul 03127, Korea; 5The Korean Nutrition Society, Seoul 04376, Korea

**Keywords:** anthocyanins, RCT, systematic review, meta-analysis, obesity criteria, inflammatory biomarkers

## Abstract

Anthocyanins, water-soluble flavonoids that produce red-to-blue pigment in plants, have antioxidant properties and have been developed as a functional food to fight obesity. In randomized controlled trials (RCTs), a systematic review with meta-analysis (SR-MA) was used to investigate these anti-obesity effects. Using search engines (PubMed, EMBASE, Cochrane-library, and CINAHL) and keywords (anthocyanins, BMI, WC, WHR, and inflammatory biomarkers), 11 out of 642 RCTs (28.3–500 mg/day of anthocyanins for 4 to 24 weeks) were included. The results showed a significant reduction in body mass index (BMI) (MD = −0.36, 95% CI = −0.58 to −0.13), but body weight (BW) and waist circumference (WC) did not change. Anthocyanins decreased BMI in the non-obese (non-OB) group in five RCTs (BMI ≤ 25; MD = −0.40 kg/m^2^; 95% CI = −0.64 to −0.16;) but did not affect BMI in the obese (OB) group. A subgroup analysis of six RCTs showed that fewer than 300 mg/day reduced BMI (MD = −0.37; 95% CI = −0.06 to −0.14), but ≥300 mg/day did not. A treatment duration of four weeks for four RCTs was sufficient to decrease the BMI (MD = −0.41; 95% CI = −0.66 to −0.16) as opposed to a longer treatment (6–8 or ≥12 weeks). An analysis of the effect of anthocyanins on the BMI showed a significant fall among those from the Middle East compared to those from Asia, Europe, South America, or Oceania. In conclusion, the anthocyanin supplementation of 300 mg/day or less for four weeks was sufficient to reduce the BMI and BW compared to the higher-dose and longer-treatment RCTs. However, further studies might be conducted regarding the dose- or period-dependent responses on various obese biomarkers.

## 1. Introduction

Anthocyanin is a natural soluble pigment in the flavonoid group. Approximately 635 or more structures have been found in nature and more than 30 types of anthocyanins and anthocyanidins have been identified based on the number and position of the hydroxyl group [1,2]. The common aglycones are pelargonidin (Pg), cyanidin (Cy), peonidin (Pn), delphinidin (Dp), petunidin (Pt), and malvidin (Mv), but Cy-3-glucoside was widely distributed [3,4]. Anthocyanins are mostly absorbed through the gastric wall with absorption rates of 10–22%, depending on chemical structure, and the bioavailability is approximately 0.26–1.8% [5,6,7]. Anthocyanins produce antimicrobial, antioxidant, and anti-inflammatory effects and play a role in the prevention and treatment of numerous chronic conditions, such as obesity, diabetes mellitus (DM), cardiovascular disease (CVD), eye diseases, and in suppressing cancer cell growth [8,9,10,11,12].

The prevalence of obesity among adult South Koreans in 2018 was 35.7% [13]. With the development of several anti-obesity foods, different attempts have been made to verify the antioxidant, insulin-sensitivity, and anti-inflammatory effects of anthocyanins [14,15,16]. However, there is a considerable lack of research on obesity effects [17,18]. Although the effect on antioxidation, reduction of lipids, and CVD biomarkers could be found, the body composition of obesity biomarkers (body weight (BW), body mass index (BMI), and waist circumference (WC)) might not change. [16,19,20,21] However, in many reports, juçara berry juice (131.2 mg/day for 6 weeks) had reduced the risk of metabolic diseases, and dried purple carrots (118.5 mg/day for 4 weeks) reduced lipids, body composition, and inflammation in obese adults [22,23]. We also found that anthocyanins (31.45 mg/day for 8 weeks) had an effect on overweight/obese adults (n = 63) [15]. After 8 weeks, a black bean test group, compared to a placebo group, showed reduced arteriosclerosis indicators (total cholesterol/high-density lipoprotein cholesterol (TC/HDLc), and low-density lipoprotein cholesterol/HDLc (LDLc/HDLc)), as well as significantly lowered BW, BMI, and WC. Although we determined the positive effects of anthocyanins concerning body composition, lipid profile, and inflammation, the effects depended on the types of anthocyanin, period of use, and subject.

Therefore, this study aimed to verify the anti-obesity effects of anthocyanins through
a systematic review using meta-analysis (SR-MA)
of clinical trials that investigated anti-obesity effects (BMI, WC, waist–hip ratio (WHR)), and the adipocytokines of anthocyanin supplementation.

## 2. Materials and Methods

### 2.1. Study Design of SR-MA

Key questions and selection criteria were determined through the SR of anthocyanin supplementation on obesity criteria and biomarkers using the PICOS model. After the collection of the relevant articles, the final set of studies for the meta-analysis was selected, and the quantitative measures for assessing the quality and meta-analysis of the articles were obtained. Two reviewers (SP and MC), who hold master’s degrees in food and nutrition from Sungshin Women’s University, applied inclusion and exclusion criteria to identify the relevant studies. Any trials that were not excluded based on title and abstract were reviewed in full-text by both reviewers who double-checked the entire processes for data collection, article selection, and article quality assessment. Any discrepancies between reviewers at any step were discussed, and a third, independent reviewer (ML), a professor of food nutrition, resolved them.

### 2.2. Data Collection

The following search engines were used: PubMed, EMBASE, Cochrane Central Control of Trials, and CINAHL, up to March 2020. The keywords for “intervention” were “anthocyanin”, “anthocyanins”, “anthocyanin-rich”, “anthocyanins rich”, “dietary anthocyanin”, “anthocyanin supplementation”, or “anthocyanin extract”; the keywords for “outcomes” were “obesity”, “body weight”, “BW”, “body mass index”, “fat mass”, “FM”, “lean mass”, “LM”, “waist circumference”, “WC”, “leptin”, “adiponectin”, “waist–hip ratio”, or “WHR”; the keywords for “study design” were “intervention”, “trial”, “randomized”, “randomized”, “random”, “randomly”, “placebo”, or “RCT”. This study was restricted to articles published in English, and the criteria for obesity were BMI, BW, WC, or WHR depending on the study goals.

### 2.3. Inclusion and Exclusion Criteria

This meta-analysis based on the PICOS design included (a) randomized controlled trials (RCTs) with anthocyanin supplements in the control group that did not receive the intervention; (b) men and women over 18; and (c) changes in BW, BMI, WHR, leptin, and adiponectin. The exclusion criteria were nonhuman subjects; non-intervention; intervention in anthocyanin deficiency; anthocyanins combined with other medications; non-RCT studies (cohort, case–control, or cross-section); review or commentary; discordance in the outcome of MetSyn; or non-English publication.

### 2.4. Quality Assessment (ROB)

The reliability of the articles in the final set of the meta-analysis was evaluated through quality assessment using the Cochrane Collaboration Risk of Bias (RoB), which combines a checklist and domain evaluation format with simplified questions. It was developed to minimize the possibility of subjective or arbitrary responses to identical questions as well as variation in assessment results due to differences in the understanding and proficiency of the reviewers concerning the methodology. The quality assessment was performed by two independent reviewers (S.P. and M.C.), while the level of RoB was determined through discussion. The third independent reviewer (M.L.) resolved any disagreements in RoB evaluation. The following seven items were examined based on the risk levels “high”, “low”, and “unclear”: Random sequence generation, allocation concealment, blinding of the participants and personnel, blinding of the outcome assessments, incomplete outcome data, selective reporting, and other sources of bias.

### 2.5. Data Analysis

The mean differences (MD) between the baseline and final value of the factors were extracted from the studies. A statistical meta-analysis was conducted using RevMan software (Version 5.3: Review Manager [RevMan], Nordic Cochrane Centre, Copenhagen, Denmark). The effect size was a quantitative measure that summarized the results, reflecting the magnitude or intensity of the relationship among the different study parameters. In this review, the effect size was based on differences in mean values as the treatment continued. The confidence interval (CI) for significance testing was 95%. For the included primary articles that did not report standard deviation (SD), the method of calculation reported by Higgins et al. was used [24]. The result of the meta-analysis was presented on forest and funnel plots [25]. The forest plot was presented in line with the statistical analysis to show the effect size (quadrangle) and 95% CI (line of symmetry). A fall in the width of the 95% CI increased the area of the quadrangle. Statistical analysis included the significance (*p* value), summary estimate (mean ± SD, 95% CI), and heterogeneity (I^2^). When the standard deviation (SD) for mean differences was not reported, it was calculated by the following formula: SD = square root ((SD baseline)^2^ + (SD final)^2^ − (2R × SD baseline × SD final)), assuming the correlation coefficient (R) = 0.5 [26]. Funnel plots were used to assess publication bias. The Higgin’s I^2^ value was used as a statistical test to verify heterogeneity [24]. In general, 0% ≤ I^2^ ≤ 25% indicated a low level of heterogeneity; 25% ≤ I^2^ ≤75% indicated a moderate level, and 75% ≤ I^2^ ≤ 100% indicated a considerably high level. Additionally, the sub-group analysis was estimated using the same methods.

## 3. Results

### 3.1. Study Selection

The search identified 1379 articles, and after the removal of duplicates, 737 were screened by title and abstract. Most (*n* = 620) were excluded because they were not relevant: Duplicates (*n* = 22), nonrandomized controlled trial (*n* = 53), and review or non-human study (*n* = 545). After assessing the full text of 22 potentially related articles, eleven articles were included. The most important reasons for exclusion were non-compliance with SR-MA criteria: No placebo group (4); subjects unrelated to the aim of the review (athletes, other functionality, age) (4); or sex bias, a short communication study, or lacked anthocyanin information (3). Figure 1 shows the study identification and selection process.

### 3.2. Characteristics of the Included RCTs

Among the 11 RCTs selected for analysis (*n* = 833), nine (*n* = 401), were included for the analysis of the correlation between BMI and anthocyanin supplementation, five (*n* = 233) for the correlation between BW and anthocyanin supplementation, and four (*n* = 148) for the correlation between WC and anthocyanin supplementation (Table 1). However, no study reported the correlation between inflammatory biomarkers and anthocyanin supplementation; hence, this was not considered in this review. The number of subjects in the 11 RCTs ranged between 16 and 80; the daily dose of anthocyanin supplementation ranged between 28.3 and 500 mg/day, and the duration of intervention ranged between 4 and 24 weeks. The countries where these were conducted were China [11], Serbia [12], Italy [14], South Korea [15], Germany [16], Italy [19], the U.K. [20], Denmark [21], Brazil [22], Australia [23], and Iran [27]. The analysis of subject characteristics showed that four RCTs did not target patients [16,19,20,21]; one RCT targeted patients with type II DM [11]; four RCTs targeted overweight or obese patients [12,14,22,23]; one RCT targeted patients with CVD [14]; and one RCT targeted patients with hyperlipidemia [27].

### 3.3. Quality Assessment; ROB

To assess the risk of bias in RCTs, the RoB tool developed by the Cochrane group was used. For random sequence generation, 10 among the 11 RCTs used computerized generation, where the RoB was found to be low [11,12,14,15,16,19,21,22,23,27], and 1 RCT did not report on the method of generation [20]. The RoB of allocation concealment was “low” in seven RCTs [14,15,16,20,21,22,23] and “unclear” in four. For the double-blinding of the participants and personnel, the RoB was “low” in nine RCTs [11,12,14,15,19,20,21,22,27], “high” in one, and “unclear” in one due to the lack of clear description [23]. For the blinding of the outcome assessments, the RoB was “low” in nine RCTs and “unclear” in two [16,27]. The RoB of the incomplete outcome data and selective reporting was “low” in 11 RCTs (Figure 2).

### 3.4. Anthocyanin Supplements on the Reduction of BMI

The analysis of the effect of anthocyanin supplementation on BMI in nine RCTs (*n* = 401) showed a reduction of 0.36 kg/m^2^ in the anthocyanin supplementation group, compared with the placebo group (95% CI = −0.58–0.13; I^2^ = 0%; *p* = 0.002) (Figure 3A). The funnel plot showed that the nine RCTs were in symmetry, indicating low publication bias. When the BMI was analyzed after categorizing the subjects into non-obese (non-OB) healthy individuals (five RCTs; BMI < 25) and obese (OB) individuals (four RCTs; BMI ≥ 25), the BMI after anthocyanin supplementation fell by 0.40 kg/m^2^ (95% CI = −0.64–0.16; I^2^ = 0%; *p* = 0.001) (Figure 3B). When the subjects were categorized into three groups according to anthocyanin dosage, supplementation BMI in Group 1 showed a significant reduction of 0.37 kg/m^2^ (six RCTs; ≤300 mg/day), compared to that of Group 2 (three RCTs; 95% CI = −0.60–−0.14; I^2^ = 0%; *p* = 0.002, Figure 3C). When the subjects were categorized into three groups according to dose duration, anthocyanin supplementation in the Group 1 BMI produced a significant reduction (four RCTs; −0.41 kg/m^2^; 4 weeks), compared to that of Group 2 (three RCTs: 6–8 weeks) and Group 3 (two RCTs; ≥12 weeks; 95% CI = −0.66–−0.16; I^2^ = 0%; *p* = 0.001, Figure 3D). However, when the intervention periods were classified by group, Group 1 (seven RCTs: 4–8 weeks) and Group 2 (two RCTs; ≥12 weeks), significant BMI reduction was found in Group 1 (−0.36 kg/m^2^; 4–8 weeks; 95% CI = −0.59–0.14; I^2^ = 0%; *p* = 0.002). 

### 3.5. Anthocyanin Supplements on the Reduction of BW or WC

The analysis of the effect of anthocyanin supplementation on BW in five RCTs (n = 233) showed that it increased by 0.15 kg, compared with that of the control group (95% CI = 0.01–0.29; I^2^ = 0%; *p* = 0.04) (Figure 4A). The funnel plot showed that the five RCTs were in symmetry, indicating a low publication bias. After categorizing the subjects into healthy (four RCTs, non-OB; BMI < 25) and obese (one RCT, OB; BMI ≥ 25) individuals, the BW was higher by 0.15 kg in the former after anthocyanin supplementation (95% CI = 0.01–0.29; I^2^ = 0%; *p* = 0.03) (Figure 4B). When the subjects were categorized into two groups according to dosage, the BMI was higher by 0.15 kg in Group 1 (four RCTs; ≤300 mg/day) than in Group 2 (one RCT; >300 mg/day; 95% CI = 0.01–0.29; I^2^ = 0%; *p* = 0.04) (Figure 4C). When the subjects were categorized into two groups according to intervention duration, the BW in Group 1 (two RCTs; ≤4 weeks) was significantly higher than that in Group 2 (three RCTs; >4 weeks), and after anthocyanin supplementation, the BW was higher by 0.15 kg in the former (95% CI = 0.01–0.29; I^2^ = 0%; *p* = 0.04, Figure 4D).

The effect of anthocyanin supplementation on WC in four RCTs (n = 148) showed no difference to that of the placebo group (*p* = 0.49). The funnel plot shows that the four RCTs were in symmetry, which indicated low publication bias.

### 
3.6. Anthocyanins Supplements on BMI and BW by National Origin


The subjects were categorized into five groups according to geography: Group 1 (four RCTs, Europe), Group 2 (two RCTs, Asia (China, South Korea)), Group 3 (one RCT, Middle East (Iran)), Group 4 (one RCT, South America (Brazil)), and Group 5 (one RCT; Oceania (Australia)). The analysis of the effect on BMI showed a significant fall in Group 3 (Figure 5A). When the subjects were categorized into two group, Group 1 (one RCT; Asian, South Korea) and Group 2 (four RCTs; Caucasian, Europe), anthocyanin supplementation significantly increased the BW by 0.15 kg in Group 2 (95% CI = 0.01–0.29; I^2^ = 0%; *p* = 0.03) (Figure 5B). In summary, this SR-MA showed that while there was a significant reduction in BMI, there was no reduction in BW or WC, and ethnic difference influenced each indicator of obesity (Table 2).

## 4. Discussion

The previous SR-MAs for anthocyanins reported on dyslipidemia [28], vascular inflammation [29], lipid composition and inflammation [30], heart disease [31], hypertension [32,33,34], DM [35], and antioxidation [36]. Thus far, only a few have been conducted on obesity-related studies for the anti-obesity effect of a flavanol complex [36] and the weight-reducing effect of resveratrol (400–800 mg/day, 1–24 weeks) [37]. In this SR-MA, we excluded the anthocyanin complex or other variants for the anti-obesity effects of anthocyanin. A notable advantage of the SR-MA is that the risk of error is reduced, which increases the reliability as the overall sample size is increased by compiling different small-scale studies. The anti-obesity effects of various anthocyanins reviewed in this analysis are thus expected to contribute to the generalization.

When the total subjects in 11 RCTs (*n* = 833) were categorized into healthy individuals (five RCTs, non-OB; BMI < 25) and obese individuals (four RCTs, OB; BMI ≥ 25), the BMI of healthy individuals was significantly reduced compared to that of the OB group. In the SR-MA anti-obesity effect of flavonols (58 studies), the BMI was reduced by 0.28 kg/m^2^ in the flavanol-intake group, and it was reduced in group with BMI ≥ 25 more than the control [36]. Our study found that anthocyanins had a stronger effect in healthy individuals compared to those who were obese. The risk factors of diseases might not have affected the results because CVD, hyperlipidemia, or DM patients were included in both the non-OB and OB groups. In other RCT studies, the anthocyanin supplementation in chokeberry juice (28.5 mg/day) [12], blueberries (375 mg/day) [14], and *Vaccinium* extract (180 mg/day) [27] for 4–6 weeks had protective effects on oxidative stress, dyslipidemia, and inflammatory obesity in CVD patients. The refined anthocyanin intake of blueberry (320 mg/day) for 24 weeks reduced dyslipidemia and insulin resistance in DM patients (n = 58) [11]. These findings indicated that anthocyanin supplementation had a beneficial effect in patients with OB, CVD, and DM. Thus, the substantial BMI reduction in the non-OB group after supplementation suggests that anthocyanins can prevent obesity.

In this SR-MA, BMI was significantly reduced in the low-dose group (seven RCTs; ≤ 300 mg/day) and short-dose period (seven RCTs; 4–8 weeks) compared to the group with high-intake (two RCTs for 300 and 500 mg/day) or longer duration (two RCTs for 12 and 24 weeks). It was hard to draw a conclusion when only two studies had doses higher that 300 mg/ day and ran longer than 12 weeks. The ethnic difference according to anthocyanin supplementation was such that the BMI was reduced in individuals from the Middle East, but since only a single RCT was assessed, the finding remains inconclusive. The effect on WC was not significant in this study, which was consistent with the results of other studies [38]. The BW also increased in the non-OB group (four RCTs; BMI < 25) that had a dose of ≥300 mg/day within 4 weeks compared to the control. A significant increase in BW was observed among Europeans, whereas the BW among Koreans decreased, but as only a single RCT was assessed, a definite conclusion could not be made.

As collectively suggested by this SR-MA, a positive effect on BMI and BW reduction was anticipated for anthocyanin supplementation of ≥300 mg/day for 4 weeks, so further studies should be conducted regarding the dosage response. It was extremely difficult to estimate the daily intake or recommended level of anthocyanins due to large differences across different countries (Korea, 3.3–95.5 mg/day; U.S., 12.5 mg/day; and Europe, 30 mg/day) and individual dietary patterns [3,7,35]. Considering the mean absorption rate of 10–20%, approximately 30 mg/day upon the intake of 300 mg/day might be assumed to have a positive effect on human metabolism. Moreover, when we applied the results of anthocyanin RCTs to those who wanted to reduce BMI, we recommended minimum levels over shorter periods to reduce any risk of obesity.

The limitations of this study are as follows: First, there was no study of the literature search on the biomarkers of obesity, so the SR-MA only analyzed the correlation between obesity indicators and anthocyanin supplementation. Second, the subjects in this study included patients with DM, obesity, dyslipidemia, and CVD, which implied a potential influence on the BMI, BW, and WC. Furthermore, the disease factor made identification of the dose-dependency effect on BMI or BW reduction more difficult. Third, although the anthocyanin studies varied according to dosage, duration, and national origin, the number of RCTs (or the number of subjects) was too low to analyze the clinical effects since there has been a general lack of study on absorption rates according to ethnicity or anthocyanin polysaccharide structures. This review primarily investigated the effect of anthocyanin supplementation on obesity indicators, not on lipid composition, which has been frequently published. This study might be included in future assessments to provide comprehensive data on the preventive effect of anthocyanin on obesity.

## 5. Conclusions

We found that the anthocyanin supplementation of 300 mg/day or less for 4 weeks was sufficient to reduce BMI and BW compared to the results from higher-dose and longer-treatment RCTs. Considering the daily intake, types of supplementation, and absorption rate, further RCTs should be conducted regarding the dose- or period-dependency responses on various obese biomarkers, such as adipocytokines.

## Figures and Tables

**Figure 1 nutrients-13-02121-f001:**
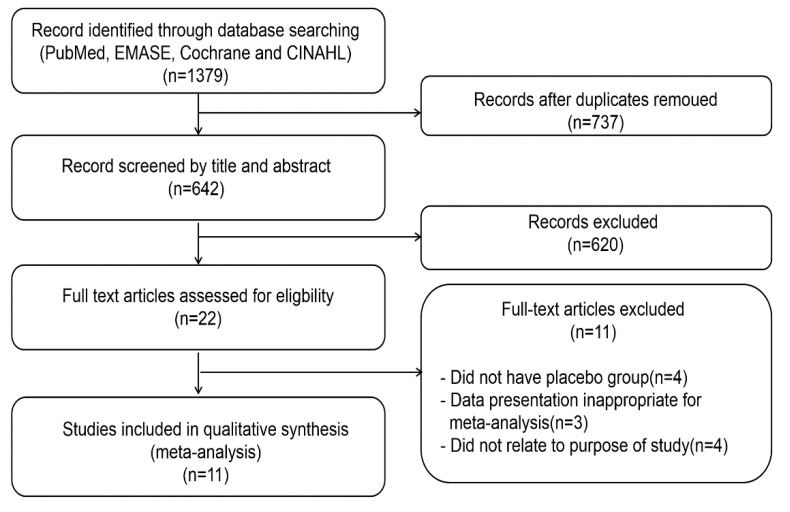
Flow chart of the study selection process for SR-MA.

**Figure 2 nutrients-13-02121-f002:**
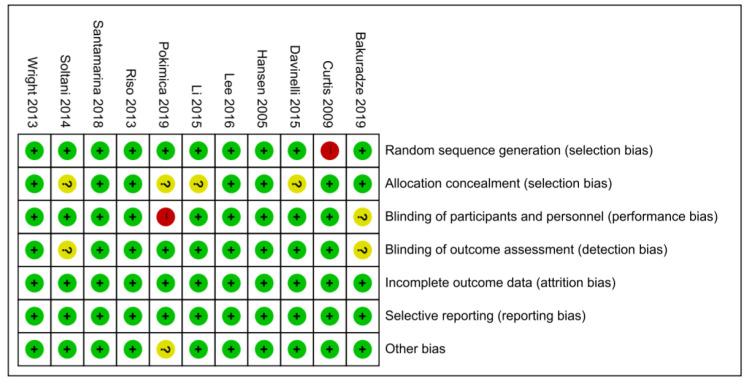
Risk of bias (ROB) results of quality assessment of the 11 RCTs. Risk of bias levels; low (green or “+”), Unclear (yellow or “?”), High (red or “-“)

**Figure 3 nutrients-13-02121-f003:**
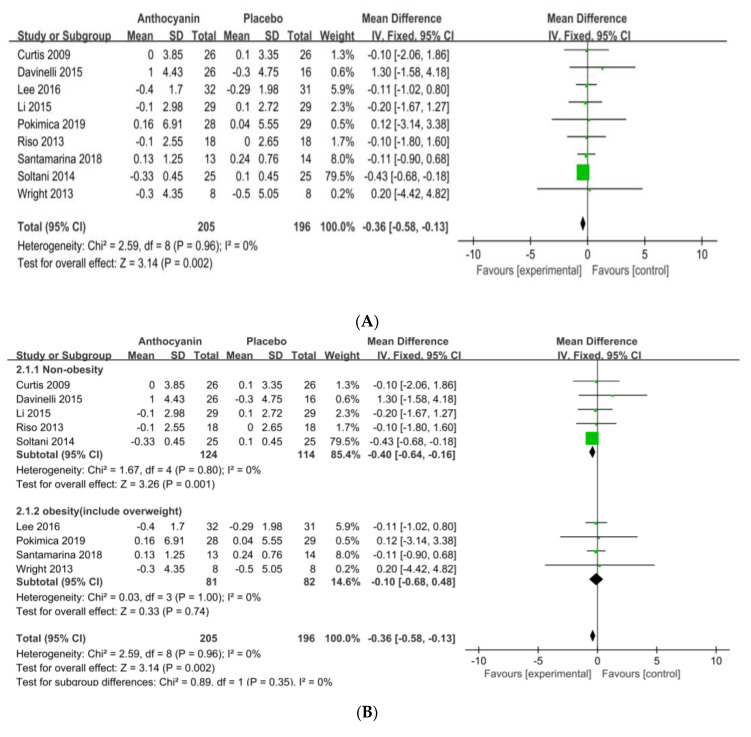
Forest plot of BMI changes (**A**) and subgroup analysis by obesity degree (**B**), intervention levels (**C**), and intervention periods (**D**). Abbreviations: SD: standard deviation, CI: confidence interval, df:degrees of freedom, Statistical heterogeneity was assessed by I^2^ test, Chochran's Q-test; I^2^: Higgin's I^2^ test, Chi^2^: Cochran’s Q-test, IV: inverse variance, Z: Z-score, P: P-value.

**Figure 4 nutrients-13-02121-f004:**
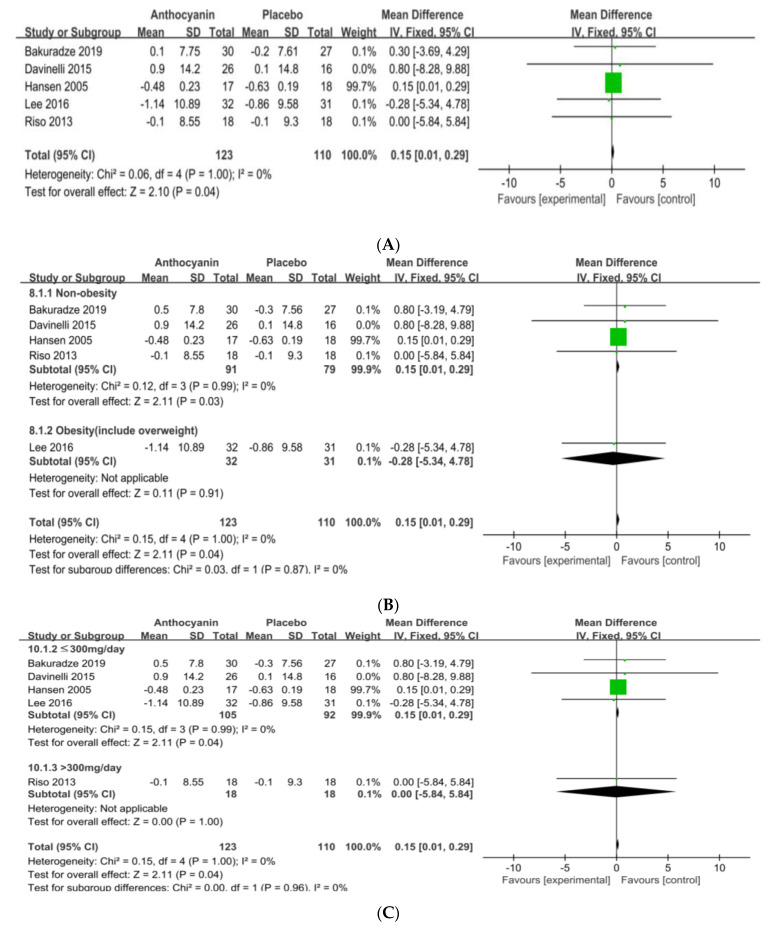
Forest plot of BW changes (**A**) and the subgroup analysis of obesity degree (**B**), treated levels (**C**), and intervention periods (**D**). Abbreviations: SD: standard deviation, CI: confidence interval, df:degrees of freedom, Statistical heterogeneity was assessed by I^2^ test, Chochran's Q-test; I^2^: Higgin's I^2^ test, Chi^2^: Cochran’s Q-test, IV: inverse variance, Z: Z-score, P: P-value.

**Figure 5 nutrients-13-02121-f005:**
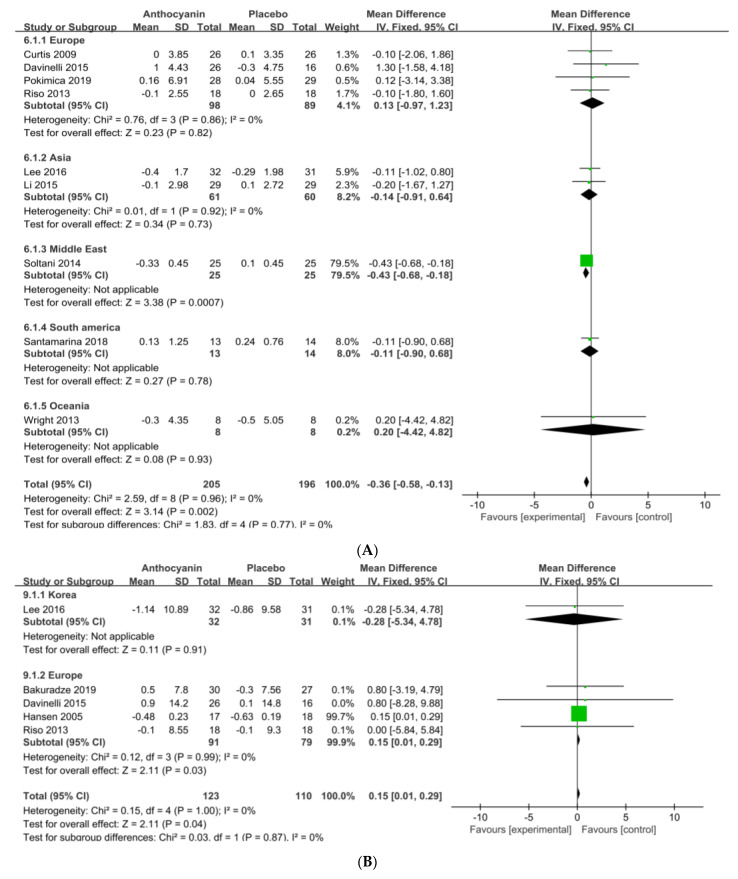
Forest plot of the changes of BMI (**A**) and BW (**B**) by country. Abbreviations: SD: standard deviation, CI: confidence interval, df:degrees of freedom, Statistical heterogeneity was assessed by I^2^ test, Chochran’s Q-test; I^2^: Higgin's I^2^ test, Chi^2^: Cochran's Q-test, IV: inverse variance, Z: Z-score, P: P-value.

**Table 1 nutrients-13-02121-t001:** Characteristics of included studies.

Study		Population					Intervention		Bio-Markers of Obesity
Author (Country)	Year	Case	Mean Age (int/cont) ^(1)^	Sex (M/F) ^(2)^	BMI (int/cont)	Sample Size (int/cont)	Duration (weeks)	Dose (mg/day)	
Bakuradze et al. [16] (Germany)	2019	Healthy male volunteers	23/24	57/0	23.0/23.0	30/27	8	205.875 (anthocyanin fruit juice)	BW
Curtis et al. [20] (U.K)	2009	Healthy postmenopausal women	58.1/58.3	0/52	25.1/24.3	26/26	12	500 (elderberry extract)	BMI
Davinelli et al. [19] (Italy)	2015	Healthy	55/55	29/13	28.9/28.5	26/16	4	162 (maqui berry extract)	BW BMI WC
Hansen et al. [21] (Denmark)	2005	Healthy	51/5	16/19	25.8/24.6	17/18	4	M: 71/F: 48 (red grape extract)	BW
Lee et al. [15] (South Korea)	2016	Overweight, obese Korean Adults	30.88/30.30	50/30	27.74/28.00	32/31	8	31.45 (black soybean extract)	BW BMI WC
Li et al. [11] (China)	2015	T2DM	58.1/57.6	34/24	24.2/23.9	29/29	24	320 (mixture of blueberry and black currant)	BMI
Pokimica et al. [12] (Serbia)	2019	Overweight or obese people	40.6/40.6	NA	26.59/27.29	28/29	4	28.3 (choke berry juice)	BMI
Riso et al. [14] (Italy)	2013	CVD	47.8	18/0	24.9/24.9	18/18 (cross-over)	6	375 (blueberry juice)	BW BMI
Santamarina et al. [22] (Brazil)	2018	Obese Adults	44.26/45.07	11/16	34.63/33.82	13/14	6	131.2 (Juçara berry freeze-dried pulp)	BMI WC
Soltani et al. [25] (Iran)	2014	Hyperlipidemia	47.93/46.36	20/30	25.40/25.21	25/25	4	180 (Vaccinium extract)	BMI
Wright et al. [23] (Australia)	2013	Overweight and obese Adults	51.4/55.0	16/0	32.4/34.0	8/8	4	118.5 (Dried purple carrot)	BMI WC

^(1)^ int: Intervention, cont: Control. ^(2)^ M: Male, F: Female. T2DM: Type 2 diabetes mellitus, CVD: Cardiovascular disease, WC: Waist circumference, BW: Body weight, BMI: Body mass index.

**Table 2 nutrients-13-02121-t002:** Subgroup analysis by biomarker condition.

		Quantitative Synthesis of Data					Heterogeneity of Data	
Biomarkers		No. of RCTs	MD	95% CI	Z-Value	*p*-Value	I^2^	*p*-Value
BMI		9	−0.36	−0.58, −0.13	3.14	0.002	0%	0.96
	Non-obese	5	−0.40	−0.64, −0.16	3.26	0.001	0%	0.80
	Obese	4	−0.10	−0.68, 0.48	0.33	0.74	0%	1.00
	≤300 mg/day	6	−0.37	−0.60, −0.14	3.15	0.002	0%	0.79
	>300 mg/day	3	−0.14	−1.11, 0.82	0.29	0.77	0%	0.99
	≤4 weeks	4	−0.41	−0.66, −0.16	3.27	0.001	0%	0.67
	6–8 weeks	3	−0.11	−0.67, 0.45	0.38	0.70	0%	1.00
	≥12 weeks	2	−0.16	−1.34, 1.01	0.27	0.78	0%	0.94
	Europe	4	0.13	−0.97, 1.23	0.23	0.82	0%	0.86
	Asia	2	−0.14	−0.91, 0.64	0.34	0.73	0%	0.92
	Middle East	1	−0.43	−0.68, −0.18	3.38	0.0007	-	-
	South America	1	−0.11	−0.90, 0.68	0.27	0.78	-	-
	Oceania	1	0.20	−4.42, 4.82	0.08	0.93	-	-
BW		5	0.15	0.01, 0.29	2.10	0.04	0%	1.00
	Non-obese	4	0.15	0.01, 0.29	2.11	0.03	0%	0.99
	Obese	1	−0.28	−5.34, 4.78	0.11	0.91	-	-
	≤300 mg/d	4	0.15	0.01, 0.29	2.11	0.04	0%	0.99
	>300 mg/d	1	0.00	−5.84, 5.84	0.00	1.00	-	-
	≤4 weeks	2	0.15	0.01, 0.29	2.10	0.04	0%	0.89
	>4 weeks	3	0.30	−2.46, 3.06	0.21	0.83	0%	0.94
	South Korea	1	−0.28	−5.34, 4.78	0.11	0.91	-	-
	Europe	4	0.15	0.01, 0.29	2.11	0.03	0%	0.99
WC		4	0.55	−0.01, 2.11	0.69	0.49	0%	0.57

**Abbreviation;** BMI: Body mass index, BW: Body weight, WC: Waist circumference, RCT: Randomized controlled trials, MD: Mean difference, CI: Confidence interval, I^2^: Higgin’s I^2^ test.

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
