# Peer review of "Effects of Anthocyanin Supplementation on Reduction of Obesity Criteria: A Systematic Review and Meta-Analysis of Randomized Controlled Trials"

_nutrients, 2021, doi:10.3390/nu13062121_

Round 1

Reviewer 1 Report

The manuscript by Park et al. examines the effects of anthocyanin supplementation on obesity with a systematic review and meta-analysis. The information generated is interesting and helpful for those in the field of study. Below are the comments:

  1. A major problem with this manuscript is the English grammar and sentence structure. The manuscript needs to be edited by a qualified person or professional English editing service. It is not suitable for publication the way it is written.
  2. The authors should define the abbreviations (e.g., BMI, WC, WHR, RCT etc.) when first used in the text. This will help the reader better understand what is being examined.
  3. The resolution for Table 1 needs to be improved. The image is not very clear.
  4. In the Conclusion section (Line 313) the authors state that a positive effect on BMI and BW reduction is anticipated for anthocyanin supplementation of ≥300mg/day dosage. Data show a decrease in BMI and BW with a dosage ≤300mg/day. Please clarify since the impression is less anthocyanin and shorter time of treatment results in decreased BMI and BW.
  5. In the Discussion section the authors should add some statements regarding the significant findings: Why less anthocyanin concentration and shorter period of treatment has greater effect than more anthocyanin and longer period? Why was the effect on BMI significant in the Middle East group compared to others?

Author Response

Reviewer 1

The manuscript by Park et al. examines the effects of anthocyanin supplementation on obesity with a systematic review and meta-analysis. The information generated is interesting and helpful for those in the field of study. Below are the comments:

  1. A major problem with this manuscript is the English grammar and sentence structure. The manuscript needs to be edited by a qualified person or professional English editing service. It is not suitable for publication the way it is written.

ANS> This paper has been edited by English Editing Services, Editage (#EO JHB_12) before it was submitted. However, it was edited by MDPI English Editing services again. (English editing ID: english-30994)

  1. The authors should define the abbreviations (e.g., BMI, WC, WHR, RCT etc.) when first used in the text. This will help the reader better understand what is being examined.

ANS> Corrected

      3. The resolution for Table 1 needs to be improved. The image is not very clear.

ANS> Corrected

     4. In the Conclusion section (Line 311) the authors state that a positive effect on BMI and BW reduction is anticipated for anthocyanin supplementation of ≥300mg/day dosage. Data show a decrease in BMI and BW with a dosage ≤300mg/day. Please clarify since the impression is less anthocyanin and shorter time of treatment results in decreased BMI and BW.

ANS> Corrected (line 304)

     5. In the Discussion section the authors should add some statements regarding the significant findings: Why less anthocyanin concentration and shorter period of treatment has greater effect than more anthocyanin and longer period? Why was the effect on BMI significant in the Middle East group compared to others?

ANS> According to the significance of subgroup-analysis (Fig 3-c&d), we concluded that anthocyanin treatment of ≤300mg/d for 4 weeks had been shown the reduction of BMI and BW. (While revision, we re-analyzed and corrected them) In these cases, since they mostly used anthocyanin extracts from the food sources in RCT intervention, their effect on BMI reduction was efficient rather than other types. However, higher than 300mg/d for longer than 12 weeks (2RCTs) did not reduce BMI or BW. In my opinion, a higher intake of anthocyanin (>300mg/d) is actually accompanied by a high dietary intake a day, particularly, for long-term periods. Another reason is that the plan of higher 300 mg/d with anthocyanin extracts is hard to conduct for longer than 12 weeks in the human according to the standard RCT designs (12 weeks, randomized, double-blind & placebo-controlled study). Generally, the purpose of longer treated study was to improve the risk of diseases such as CVD, hyperlipidemia and so on. [Ex; Nut, Met, & Cad Dis, 2013,23;843] However, when we apply the results of anthocyanin RCTs in the population who want to reduce the risk of diseases, we have to recommend the minimum levels over shorter periods to reduce any risk of obesity. Therefore, it was very difficult to make a conclusion based on the few RCTs for higher & longer treatment of anthocyanin at this time.

I agree with you, it was difficult to say the anthocyanin’s effect on BMI reduction in the Middle East group according to one RCT evidence. However, the four other groups, EU, Asia, South America, and Oceania, were not significant on the effects. According to our subgroup analysis, we carefully suggest that anthocyanin’s on BMI or BW reduction may be race-dependent, and you may consider it for future study. That is why this race–dependent result was excluded in “Conclusion”.

Reviewer 2 Report

The authors of the systematic review entitled “Effects of anthocyanin supplementation on obesity: a systematic review and meta-analysis of randomized controlled trials” investigated, by a meta-analysis, the potential anti-obesity effects mediated by anthocyanins.

Comments:

Please define acronyms the first time that they appear in the main text.

Line 49. BW, BMI, and WC.

Line 55. TC, HDLc, and LDLc.

Line 61. SR-MA

In the flow chart, the first exclusion step was applied due to duplicates articles. In the second exclusion (n=620), a reason for exclusion is to duplicate articles. This should not be in the first exclusion? It is not clear.

The table 1 and 2 must be editable text, not an image.

Figures 2A and 2B give the same information. I consider that figure 2B is the most appropriate (the most complete).

Author Response

Reviewer 2

The authors of the systematic review entitled “Effects of anthocyanin supplementation on obesity: a systematic review and meta-analysis of randomized controlled trials” investigated, by a meta-analysis, the potential anti-obesity effects mediated by anthocyanins.

Comments:

  1. Please define acronyms the first time that they appear in the main text.

Line 49. BW, BMI, and WC/ Line 55. TC, HDLc, and LDLc/ Line 61. SR-MA

ANS> Corrected

  1. In the flow chart, the first exclusion step was applied due to duplicates articles. In the second exclusion (n=620), a reason for exclusion is to duplicate articles. This should not be in the first exclusion? It is not clear.

ANS> 1st exclusion was done by data engines of Pubmed, EMBASE, Cochrane library, and CINAHL to exclude duplicates. However, we need 2nd exclusion for duplication (n=22), nonrandomized RCT (n=53), review or nonhuman (n=545 ) after abstracts of manuscripts were carefully observed. We revised Fig 1.

  1. The table 1 and 2 must be editable text, not an image.

ANS> Corrected

  1. Figures 2A and 2B give the same information. I consider that figure 2B is the most appropriate (the most complete).

ANS>  Although Fig 2A is described as the combined result of 11 RCTs according to 7 criteria of ROB, as all 11 RCTs were used, Fig 2B can explain enough.

Reviewer 3 Report

Well-written review which sums up the information as to the duration of the supplementation and dosage.

abbreviations should be explained in the place where they appear in the text for the first time,
The language should be checked by the native speaker,
Abstract:

the title is "Effects of anthocyanin supplementation on obesity"
the conclusion is: In conclusion, the effects of antho-25 cyanins on BMI were depended on the obesity-degree, dose-levels, dose-period, and ethnics differ-26 ences.
this is misleading obesity vs. BMI, obesity is more than just a BMI, obesity is related to vascular disorders, oxidative stress, body weight gain. Thus the conclusion is not related to the title. I would expect more than BMI after reading the manuscript. This is why the quality of the data is AVERAGE.

Table 1 looks like a copy from a book but not the genuine one. Must be improved.

Discussion:
resveratrol is not anthocyanin, instead, berries should be discussed
discussion is blurred and difficult to follow
in the title should signify that this is a study on humans
The English language needs huge correction

Conclusions:
again is not adequate with the article title: "Effects of anthocyanin supplementation on obesity", no: rather should be ON THE BMI. It will be more interesting to analyze other metabolic disorders in obesity + anthocyanins.

the only conclusion is that:
BMI and BW reduction is anticipated for the anthocyanin supplementation of ≥300 mg/day dosage and 4–12 week duration.
this is a poor conclusion for a researcher who is doing experiments: Rather it will be more interesting to find the difference between 4 weeks vs. 8 weeks and 12 weeks of the supplementation. There is a huge difference between these 3 supplementation periods.

In conclusion, the title is misleading and the results are of a little quality in that form. It may not get a high impact on the scientific community in that form. 

Author Response

Reviewer 3

Well-written review which sums up the information as to the duration of the supplementation and dosage.

  1. abbreviations should be explained in the place where they appear in the text for the first time,

ANS> Corrected

  1. The language should be checked by the native speaker,

ANS> This paper has been edited by English Editing Services, editage (#EO JHB_12) before it was submitted. However, it was edited by MDPI English Editing services again. (English editing ID: english-30994)

  1. Abstract: the title is "Effects of anthocyanin supplementation on obesity"
    the conclusion is: In conclusion, the effects of anthocyanins on BMI reduction were depended on the obesity-degree, dose-levels, dose-period, and ethnics differences.
    This is misleading obesity vs. BMI, obesity is more than just a BMI, obesity is related to vascular disorders, oxidative stress, body weight gain. Thus the conclusion is not related to the title. I would expect more than BMI after reading the manuscript. This is why the quality of the data is AVERAGE.

ANS> For the data collection, the PICOS design model was conducted for SR-MA. For Intervention, outcome, and study design, we used various appropriate keywords. (Line 82-92) We used BMI, BW, WC, LBM, FM, etc as keywords of obesity criteria, and leptin and adiponectin as keywords of obesity biomarkers. Eventually, 11 RCTs were screened, and unscreened RCTs were automatically excluded in SR-MA such as adipokines for obesity biomarkers, unfortunately. Moreover, we also excluded RCT treated with the complex including anthocyanin, and RCTs to find lipid profiles as biomarkers, which were numerously published.

Although the RCT studies including one CVD, one hyperlipidemia, and one DM patient were not excluded in 11 RCTs, they were screened by the data selection of obesity outcomes, not the other outcomes. Otherwise, it is also impossible to subgroups-meta analysis because of a single paper with CVD, HL, or DM. (See 2nd limitation; Line 293) Therefore, we need the new PICOS model for the new outcome, such as vascular disease, oxidative stress, and BW gain, and the different results of SR-MA from differently-selected RCTs were given. 

I thought the quality of the SR-MA study for RCTs was only determined by ROB methods, which was shown in Fig2. Since three authors decided ROB result including two “Unclear” and nine “Low-risk” RCTs was fine to be able to analyze Meta, and we found the anthocyanin’s effect on BMI reduction because the 81.8%, (9/11) of data have high-qualified data with “LOW-RISK”.

  1. Table 1 looks like a copy from a book but not the genuine one. Must be improved.

ANS> Corrected

  1. Discussion:
    resveratrol is not anthocyanin, instead, berries should be discussed
    discussion is blurred and difficult to follow

ANS> Cyanidin-3-glucoside is the most frequent anthocyanin found in raspberries, blackberries, elderberries, purple corn, or black carrots. Moreover, malvidin-3-glucoside is the major anthocyanin in red grapes and wines whilst pelargonidin-3-glucoside in strawberries. Comparison among various sources of anthocyanins including red grapes may be helpful to understand, however, we revised both parts of ”resveratrol” and “berries”. (line 258-262)

  1. in the title should signify that this is a study on humans
    The English language needs huge correction

ANS> I assumed “randomized controlled trials (RCT)” in the title to be a human study, since the global standard for animal RCT has not been established so far.

This paper has been edited by English Editing Services, Editage (#EO JHB_12) before it was submitted. However, it was edited by MDPI English Editing services again. (English editing ID: english-30994)

  1. Conclusions:
    again is not adequate with the article title: "Effects of anthocyanin supplementation on obesity", no: rather should be ON THE BMI. It will be more interesting to analyze other metabolic disorders in obesity + anthocyanins.

ANS> See the answers to the #3 query. We excluded many RCTs that you are interested in because many anthocyanin RCTs for metabolic biomarkers of obesity were already published. It is better to change “Effects of anthocyanin supplementation on the reduction of obesity criteria", because not only BMI but also BW were reduced by anthocyanins. However, the MDPI English services suggested “ Effects of anthocyanin supplementation on reducing obesity".

  1. The only conclusion is that: BMI and BW reduction is anticipated for the anthocyanin supplementation of ≥300 mg/day dosage and 4–12 week duration. This is a poor conclusion for a researcher who is doing experiments: Rather it will be more interesting to find the difference between 4 weeks vs. 8 weeks and 12 weeks of the supplementation. There is a huge difference between these 3 supplementation periods.

ANS> I appreciate your suggestion. After we re-analyzed among 3 periods groups, there was different among 4 (p=0.001), 6- 8 (p=0.70) & ≥12 weeks (p=0.78) with only significance of 4 weeks. (line 190-194 & Fig 3-d) Last time, since the significance between 2 groups, 4-8 weeks (p=0.002) and ≥12weeks (p=0.78) were found, we decided that the duration for BMI reduction was classified by shorter than 12 weeks and 12 weeks or more, according to the standard of RCT for 12 weeks, practically. However, we revised this.

I have been concerned that why less anthocyanin concentration and shorter period of treatment has a greater effect than more anthocyanin and longer period?  I could not mention in manuscript without the evidence, however, in my opinion; firstly, anthocyanin extracts from the food sources were mostly used in RCT with <300mg/d for a shorter period, so, their effect on BMI reduction may be efficient rather than the complex types. Secondly, a higher intake of anthocyanin (≥300mg/d) is accompanied by the surplus meals a day, and it's getting worse in the case of the long-term periods.

  1. In conclusion, the title is misleading and the results are of a little quality in that form. It may not get a high impact on the scientific community in that form. 

ANS> See the answers to the #3 query.

The # 15 reference was my RCT study that performed by 31.45mg/d anthocyanin for eight weeks, and we found significant effects on the reduction of WT, BMI, BW, and lipid profiles in obese adults compared to the placebo. Before the RCT study, the minimum levels of anthocyanin extracts for humans were calculated from the animal study. I wonder how different results come up with anthocyanin RCTs in the races with different levels and periods. However, with only a few RCTs for a high-dose and longer duration, it was inconclusive.

Moreover, when we apply the results of anthocyanin RCTs in the population who want to reduce BMI, we have policy that the minimum levels over shorter periods to provide maximum effects should be recommended to reduce any risk of obesity.

Round 2

Reviewer 1 Report

The authors did a good job with the revision.